# South-to-north migration preceded the advent of intensive farming in the Maya region

Douglas J. Kennett [1✉], Mark Lipson [2,3✉], Keith M. Prufer [4,5✉], David Mora-Marín [6], Richard J. George[1], Nadin Rohland[2], Mark Robinson[7], Willa R. Trask[8], Heather H. J. Edgar [4], Ethan C. Hill[4], Erin E. Ray [4], Paige Lynch[4], Emily Moes [4], Lexi O'Donnell [9], Thomas K. Harper[10], Emily J. Kate [11], Josue Ramos[12], John Morris[12], Said M. Gutierrez [13], Timothy M. Ryan[10], Brendan J. Culleton[14], Jaime J. Awe[12,15] & David Reich [2,3,16,17✉]

The genetic prehistory of human populations in Central America is largely unexplored leaving an important gap in our knowledge of the global expansion of humans. We report genome-wide ancient DNA data for a transect of twenty individuals from two Belize rock-shelters dating between 9,600-3,700 calibrated radiocarbon years before present (cal. BP). The oldest individuals (9,600-7,300 cal. BP) descend from an Early Holocene Native American lineage with only distant relatedness to present-day Mesoamericans, including Mayan-speaking populations. After ~5,600 cal. BP a previously unknown human dispersal from the south made a major demographic impact on the region, contributing more than 50% of the ancestry of all later individuals. This new ancestry derived from a source related to present-day Chibchan speakers living from Costa Rica to Colombia. Its arrival corresponds to the first clear evidence for forest clearing and maize horticulture in what later became the Maya region.

[1] Department of Anthropology, University of California, Santa Barbara, CA 93106, USA. [2] Department of Genetics, Harvard Medical School, Boston, MA 02115, USA. [3] Department of Human Evolutionary Biology, Harvard University, Cambridge, MA 02138, USA. [4] Department Anthropology, University of New Mexico, Albuquerque, NM 87131, USA. [5] Center for Stable Isotopes, University of New Mexico, Albuquerque, NM 87106, USA. [6] Department of Linguistics, University of North Carolina, Chapel Hill, Chapel Hill, NC 27599, USA. [7] Department of Archaeology, Exeter University, Exeter, UK. [8] Central Identification Laboratory, Defense POW/ MIA Accounting Agency, Joint Base Pearl Harbor-Hickam, Honolulu, HI 96853, USA. [9] Department of Sociology and Anthropology, University of Mississippi, University, Oxford, MS 38677, USA. [10] Department of Anthropology, The Pennsylvania State University, University Park, PA 16802, USA. [11] Vienna Institute for Archaeological Science, University of Vienna, Vienna, Austria. [12] Belize Institute of Archaeology, Belmopan, Belize. [13] Ya'axché Conservation Trust, Punta Gorda Town, Belize. [14] Institutes of Energy and the Environment, The Pennsylvania State University, University Park, PA 16802, USA. [15] Department of Anthropology, Northern Arizona University, Flagstaff, AZ 86001, USA. [16] Broad Institute of Harvard and MIT, Cambridge, MA 02142, USA. [17] Howard Hughes Medical Institute, Harvard Medical School, Boston, MA 02115, USA. ✉email: kennett@anth.ucsb.edu; mlipson@genetics.med.harvard.edu; kmp@unm.edu; reich@genetics.med.harvard.edu

Poor preservation of ancient skeletal material in the hot and humid neotropics means that little is known about Early and Middle Holocene (10,000–4000 cal. BP) population history of southeastern Mexico and northern Central America, an area that later became the Maya region. Previous ancient DNA analyses have indicated that the earliest Central and South Americans, as well as present-day groups from the same regions, descend primarily from the more southerly of two founding Native American genetic lineages[1–5]. Published early Holocene (9400–7300 cal. BP) individuals from Belize ($N = 3$) are consistent in deriving their ancestry from this same large-scale north-to-south movement of people, but they display only distant relatedness to present-day groups in Mexico and Central America, including local Maya-speaking populations[2]. Instead, Maya people today show the greatest affinities to both South Americans and Indigenous Mexicans[6,7], suggesting the potential for further episodes of population movement and admixture in this region during the past 7300 years. The genetic history of the region is essential to understand the evolution of cultures, languages, and technologies, including domesticated plant crops that transformed the neotropics.

We studied an assemblage of human remains recently excavated from two rock-shelters in Belize (Mayahak Cab Pek [MHCP] and Saki Tzul [ST] in the Bladen Nature Reserve; ~16°29′28 N, 88°54′37 W; ~430 m above sea level; Fig. 1, Supplementary Figs. 1–3)[8,9], which provide an unparalleled transect of well-preserved skeletal material spanning the past 10,000 years (Supplementary Note 1, Supplementary Figs. 4–19, Supplementary Table 1)[8]. Archeological deposits in these shelters extend back to ~12,000 cal. BP[10], including more than 50 directly radiocarbon dated individuals: 6 between 9600 and 6800 cal BP

and 47 between 5700 and 1000 cal. BP (Supplementary Note 2, Supplementary Table 2)[8]. A gap in dated human burials exists between 6800 and 5700 cal. BP. Stable isotopic dietary data ($\delta^{13}C_{collagen}$, $\delta^{15}N_{collagen}$, $\delta^{13}C_{apatite}$) from these individuals show increases in the consumption of maize starting after 4700 cal. BP[8], but it remains unclear if this dietary shift represents local adoption of more intensive maize cultivation or a new population of maize farmers moving into the region.

Maize domestication began in southwest Mexico ~9000 years ago[11,12] and genetic and microbotanical data indicate early dispersal southward and into South America prior to 7500 cal. BP[13] as a partial domesticate before the complete suite of characteristics defining it as a staple grain had fully developed[14–16]. Secondary improvement of maize occurred in South America[17], where selection led to increased cob and seed size beyond the range of wild teosinte progenitor species. Maize cultivation was widespread in northwestern Colombia and Bolivia by 7000 cal. BP[18,19]. The earliest evidence for maize as a dietary staple comes from Paredones on the north coast of Peru, where dietary isotopes from human teeth suggest maize shifted from a weaning food to staple consumption between 6000–5000 cal. BP[20] consistent with directly dated maize cobs[21]. Maize cultivation was well-established in some parts of southern Central America (e.g., Panama) by ~6200 cal. BP[22].

Starch grains and phytoliths recovered from stone tools from sites in the southeastern Yucatan near MHCP and ST indicate that maize (*Zea mays*), manioc (*Manihot* sp.) and chili peppers (*Capsicum* sp.) were being processed during the Middle Holocene, possibly as early as 6500 cal. BP[23]. Paleoecological data (pollen, phytoliths, and charcoal) from lake and wetland cores point to increases in burning and land clearance associated with

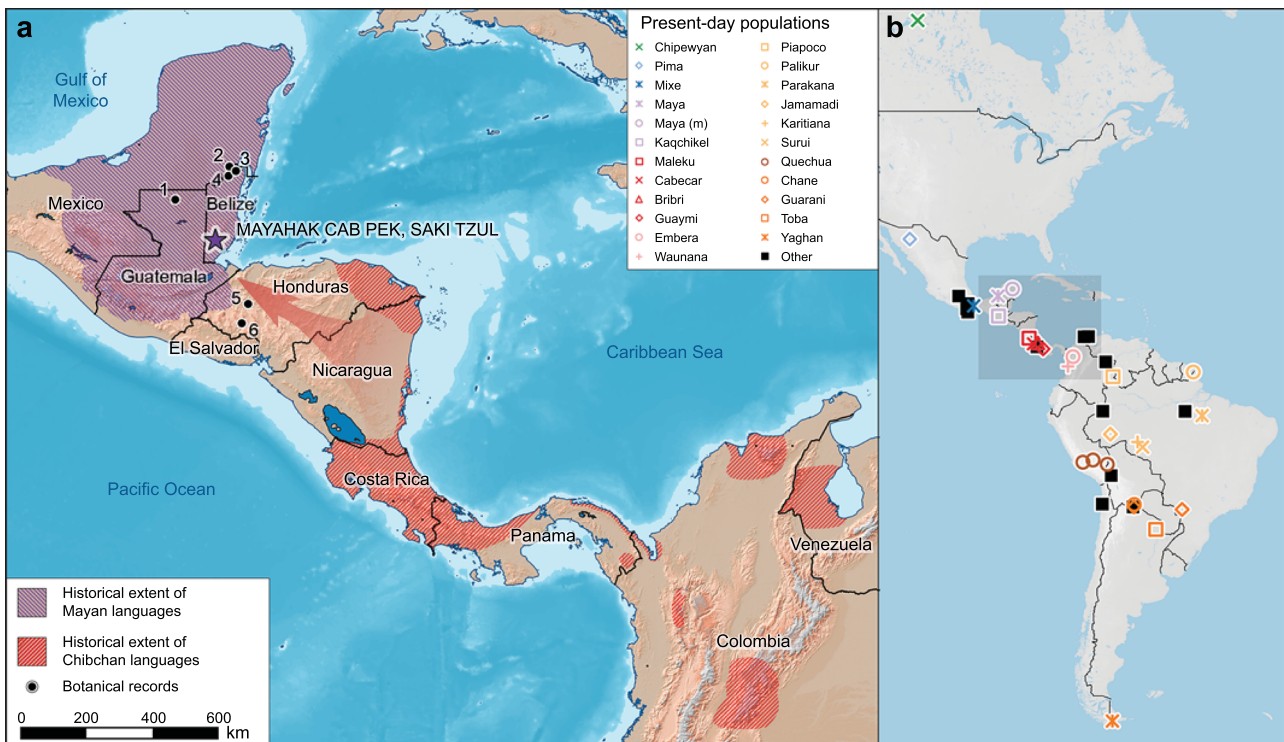

**Fig. 1 Extent of Mayan and Chibchan languages, paleobotanical records showing early horticulture, and present-day populations with genome-wide data analyzed in this study. a** Location of MHCP and ST is shown relative to the historical distributions of people speaking Mayan (purple)[46] and Chibchan (red)[45] languages. Paleo-botanical records from the southeastern Yucatan and adjacent areas with evidence of maize, manioc, and chili pepper between 6500 and 4000 cal. BP are shown in black as: 1) Lake Puerto Arturo[26]; 2) Cob Swamp[24]; 3) Rio Hondo Delta[25]; 4) Caye Coco[23]; 5) Lake Yojoa[27]; and 6) El Gigante rock-shelter[28]. The arrow shows the proposed movement of Isthmo-Colombian horticultural populations into the southeastern Yucatan by at least 5600 cal. BP. **b** Distribution of present-day groups with genome-wide comparative data (also see Supplementary Fig. 20).

early maize cultivation after ~5600 cal. BP[24,25]. Maize was grown at low levels after initial introduction, and there is greater evidence for forest clearing and maize-based horticulture after ~4700 cal. BP[26,27]. Increases in forest clearing and maize consumption coincide with the appearance of more productive varieties of maize regionally[28], which have been argued based on genetic and morphological data to to be reintroduced to Central America from South America[29].

The increases in burning, forest disturbance, and maize cultivation in southeastern Yucatan evident after ~5600 cal. BP, as well as the subsequent shift to more intensified forms of maize horticulture and consumption after ~4700 cal. BP, can be plausibly linked to: 1) the adoption of maize and other domesticates by local forager-horticulturalists, 2) the intrusion of more horticulturally-oriented populations carrying new varieties of maize, or 3) a combination of the two. Dispersals of people with domesticated plants and animals are well documented with combined archeological and genome-wide ancient DNA studies in the Near East;[30,31] Africa;[32] Europe;[33] and Central, South, and Southeast Asia[34–36]. The spread of populations practicing agriculture into the Caribbean islands from South America starting ~2500 years ago is also well documented archeologically and genetically[37,38]. However, in the American mainland, the generally accepted null hypothesis is that the spread of horticultural and later farming systems typically resulted from diffusion of crops and technologies across cultural regions rather than movement of people[39,40]. Here, we examined the mode of horticultural dispersal into the southeastern Yucatan with genome-wide data for a transect of individuals from MHCP and ST with stable isotope dietary data and direct AMS $^{14}$C dates between 9600 and 3700 cal. BP.

**Ethics and community engagement**. All ancient skeletons from the MHCP and ST rock-shelters were excavated by the Bladen Paleoindian and Archaic Archaeological Project (BPAAP) under permits issued by the Belize Institute of Archaeology (IA) and the Belize Forest Department. Skeletons of ancient individuals were exported under permits issued by the IA in accordance with the laws of Belize and permission granted to conduct molecular analyses on bulk tissues extracted from skeletons of ancient individuals. Research was conducted in close collaboration with the Ya'axché Conservation Trust, an internationally recognized Belizean NGO that is the co-manager of the Bladen Nature Reserve (BNR) with the Government of Belize. Ya'axché is locally managed and largely staffed by members of descendent Maya communities. As part of this collaboration, BPAAP research proposals are annually reviewed by the Ya'axché administrative and scientific staff. In 2016 and 2018 K.M.P. gave presentations to the local staff of Ya'axché and other interested community members on our research. In 2020, in coordination with Ya'axché, we invited indigenous leaders and community members from villages proximate to the BNR to consult on this research (K.M.P., D.J.K., and M.L. participated). We provided advance invitations and arranged transportation for forty-six people from five villages to attend the public consultation. K.M.P. delivered a presentation detailing the fieldwork, laboratory work, and results of this study, and this was followed by a session answering questions and clarifying the data and interpretations. Community members requested future public consultations to update them on additional research results, as well as copies of all study results in English with translations into the Mopan and Q'eqchi' languages. Additionally, in 2017 and 2018–2019 results of this research were presented by K.M.P. at the annual Belize Archaeology Symposium (BAS), a widely publicized venue sponsored by IA. The BAS affords the opportunity for both the presentation of research results and feedback from professional

and public communities (including members of indigenous Mayan-speaking communities).

## Results

We obtained powder from 28 skeletal samples—21 from the compact petrous region within the temporal bone, 3 from other bones, and 3 from teeth—in dedicated clean rooms (Table 1). We extracted DNA[41], and prepared Illumina sequencing libraries treated with uracil DNA glycosylase to reduce cytosine deamination associated with ancient DNA[42]. We enriched the libraries for ~1.2 million single-nucleotide polymorphism (SNP) targets[43] and sequenced each individual to between 0.002 and 3.6× average depth of coverage (Supplementary Data 1). We validated the authenticity of the ancient DNA through: 1) characteristic damage patterns at the ends of DNA fragments resulting from cytosine deamination, and 2) minimal rates of contamination as assessed by a combination of approaches (Methods; Supplementary Data 1, 2). Seven samples did not yield usable data by these criteria, and we excluded an additional four individuals with modest evidence of contamination and/or low sequencing coverage from our main analyses. Two samples were also determined to have been derived from the same individual. We determined mitochondrial (mtDNA) haplogroups for 19 of the 20 newly reported individuals and Y-chromosome haplogroups for 10 of the 13 males. Our genetic sex determinations matched the 11 available osteological assessments (Supplementary Table 1). We identified one pair of first-degree family relatives (MHCP.17.1.C1 and MCHP.17.1.1B), one pair of second-degree relatives (ST.16.1.3 and ST.16.1.2), and one pair of second/third-degree relatives (MHCP.14.1.2A and ST.16.1.1, notable for being buried at the two different rock-shelters, located ~1.5 km apart), in addition to five possible more distant relative pairs (Supplementary Fig. 21). For ten individuals with sufficiently high coverage, we inferred runs of homozygosity (ROH) using hapROH[44]. All ten display some long (>4 centiMorgans [cM]) ROH segments, indicative of small population sizes, but with a paucity of very long segments (>20 cM), suggesting avoidance of close-kin unions (Supplementary Fig. 22). Under the assumption that these individuals are representative samples, we estimated recent ancestral effective population sizes (which can be quite different from census population sizes in a given region) of 220–435 (95% confidence interval) for two early individuals and 503–811 for eight later individuals (see below).

We used principal component analysis (PCA) to visualize the genetic structure of the ancient individuals in relation to a diverse set of ancient and present-day individuals from South, Central, and North America (Fig. 2a; Supplementary Fig. 23)[6]. All individuals were projected onto axes generated using a small set of present-day populations to lessen the effects of high levels of genetic drift (Methods). The oldest ancient individuals (9600–7300 cal. BP) fall near the intersection of the main axes of variation, closest to ancient individuals from South America[2]. Individuals in the transect dating between 5600 and 3700 cal. BP shift in the direction of present-day speakers of the Chibchan family of languages[45] (hereafter Chibchan populations) from northern Colombia and Venezuela and the Isthmian region of southern Central America (see Fig. 1). Present-day speakers of the Mayan family of languages[46] (hereafter Maya populations) fall near the 5600 and 3700 cal. BP individuals but shift toward groups from western and northern Mexico. Alternative PCA plots with different populations used to compute the axes show similar patterns (Supplementary Fig. 23).

We built on these observations by using f-statistics to measure differential allele-sharing between the ancient Belize individuals and other ancient and present-day populations (Supplementary

**Table 1 Sample information.**

| Burial ID | Date (cal. BP)[b] | Sex | MtDNA | Y-chrom. | SNPs[c] |
|---|---|---|---|---|---|
| MHCP.19.12.17[ad] | 11,970–11,410[g] | M | No call | No call | 2711 |
| MHCP.17.1.8[a] | 9610–9470 | M | C1b | Q1a2a1a1 | 155274 |
| MHCP.14.1.6 | 9420–9140 | F | D4h3a5 | — | 505946 |
| MHCP.17.1.C1[a] | 8980–8590 | M | D4h3a | Q1a2a[e] | 70321 |
| MHCP.17.1.1B[a] | ~8980–8590[g] | F | D4h3a5 | — | 68002 |
| ST.16.1.3 | 7460–7320 | M | D1 | Q1a2a1a1 | 292687 |
| ST.16.1.2 | 7430–7310 | M | D1 | Q1a2a1a1 | 469913 |
| MHCP.14.1.A5[ad] | 7150–6880[g] | M | D1 | No call | 12090[f] |
| MHCP.19.12.18[a] | 5650–5490[g] | F | C1c | — | 334792 |
| MHCP.17.1.7[a] | 5590–5330 | F | C1c | — | 755033 |
| MHCP.14.1.1[a] | 5270–4880 | M | A2q | Q1a2a1a1 | 642021 |
| MHCP.14.1.5A[a] | 5220–4870 | M | A2 | Q1a2a1a1 | 1016267 |
| ST.18.11.8[a] | 5040–4860 | F | C1c | — | 126276 |
| MHCP.17.2.11A[ad] | 4970–4840 | M | A2 | No call | 34333 |
| ST.18.11.9[ad] | 4960–4820 | M | C1c | Q1a2a[e] | 10868[f] |
| MHCP.98.34.4B[a] | 4850–4650 | F | C1c | — | 679123 |
| MHCP.19.12.10[a] | 4840–4620[g] | F | C1c | — | 149758[f] |
| ST.17.7.14[a] | 4820–4520 | M | D4h3a5 | Q1a2a1a1 | 643080 |
| MHCP.14.1.2A[a] | 4790–4420 | M | A2 + (64)+@16111 | Q1a2a1a1 | 770618 |
| ST.16.1.1[a] | 4570–4420 | F | C1c | — | 826934 |
| MHCP.14.2.4A[a] | 4530–4420 | M | C1c | Q1a2a1a1 | 754676 |
| MHCP.14.2.4C[a] | 4160–3990 | M | A2 + (64)+@16111 | Q1a2a1a1 | 68261 |
| MHCP.98.34.3A[a] | 3970–3730 | M | C1c | Q1a2a1a1 | 109662 |

*MHCP* Mayahak Cab Pek, *ST* Saki Tzul.
[a]Newly reported.
[b]95.4% Confidence Interval.
[c]Unique autosomal SNP hits on the ~1.2 million target set.
[d]Excluded from genome-wide analyses.
[e]Consistent with Q1a2a1a1.
[f]Restricted to damaged sequences (see Methods).
[g]Date ranges estimated from associated radiocarbon dates on charcoal or familial relationships (see Supplementary Table 2).

Data 3). First, we computed outgroup $f_3$-statistics to determine total shared genetic drift (Supplementary Data 4). As previously observed, the individuals dating to between 9600 and 7300 cal. BP do not display a clear pattern of closer relatedness to groups from any particular geographic area, instead sharing the most drift with a range of Central and South Americans, as would be expected for an early-splitting lineage[2]. The 5600–3700 cal. BP individuals yield both higher $f_3$-statistic values and a pattern of greatest sharing with Maya and Chibchan populations.

To test directly for any asymmetrical relationships between the 9600–7300 cal. BP and 5600–3700 BP groups and present-day populations, we computed statistics of the form $f_4$(9600–7300 cal. BP, 5600–3700 cal. BP; Present-day1, Present-day2) (Supplementary Table 3; Supplementary Data 5). We observe numerous significant statistics (max $Z > 5$), with the strongest indicating excess allele-sharing between the 5600–3700 cal. BP individuals and either Chibchan populations or Kaqchikel (a Maya population from highland Guatemala) (Supplementary Table 3). We also computed pooled statistics of the form $f_4$(9600–7300 cal. BP, 5600–3700 cal. BP; Outgroup, Present-day grouping) (Fig. 2b), as well as individual-level statistics, which confirm the homogeneous 9600–7300 cal. BP and 5600–3700 cal. BP clusters observed in PCA (Supplementary Table 4; one possible but non-significant outlier in the form of the early individual MHCP.17.1.8). In addition to relatedness to Chibchan populations, the 5600–3700 cal. BP individuals display greater allele-sharing with the 9600–7300 cal. BP individuals than do present-day populations ($f_4$(Outgroup, 9600–7300 cal. BP; Present-day, 5600–3700 cal. BP) > 0, $Z > 3$; Supplementary Data 6), implying that the ancient and present-day groups cannot be related by a simple tree, and suggesting that the 5600–3700 cal. BP individuals could be admixed with ancestry related to the 9600–7300 cal. BP individuals and to ancestors of Chibchan populations.

We further examined broader patterns of allele-sharing to resolve the directionality of gene flow. If admixture occurred from a group related to the 5600–3700 cal. BP individuals into the ancestors of Chibchan populations, we would expect to find excess allele-sharing between the 9600–7300 cal. BP individuals and Chibchan populations as well, due to the relatedness between the ancient groups. However, present-day populations in Central and South America are approximately symmetrically related to the 9600–7300 cal. BP individuals ($f_4$(Outgroup, 9600–7300 cal. BP; Present-day1, Present-day2) ~0, $|Z| < 2.2$; Supplementary Data 7). By contrast, we observe significantly positive statistics $f_4$(9600–7300 cal. BP, 5600–3700 cal. BP; Present-day1, Present-day2) with several geographically and linguistically diverse South American groups in the "Present-day2" position, likely induced by shared ancestry between the South Americans and Chibchan populations (Supplementary Table 5). These signals can be explained parsimoniously by admixture from a group related to the ancestors of Chibchan populations into the ancestors of the 5600–3700 cal. BP individuals, but not in the reverse direction.

Additional evidence of admixture is provided by substantial matrilineal discontinuity between the 9600–7300 cal. BP and 5600–3700 cal. BP groups. The majority of the 9600–7300 cal. BP individuals (5 of 6, in addition to a low-coverage individual from ~7000 cal. BP) carried mtDNA haplogroup D (including D4h3a and D4h3a5 haplotypes rare in the region today), and the sixth carried haplogroup C1b. By 5600–3700 cal. BP, however, the matrilineal makeup of the sampled individuals was almost entirely different, dominated by haplogroups C1c (9 out of 15) and A2 (5 of 15), similar to distributions found broadly in Central America today[47,48]. Although changes in uniparental genotype frequencies can sometimes be driven by random genetic drift, the almost entirely different sets of mtDNA haplogroups observed between the two time periods are consistent with the evidence of

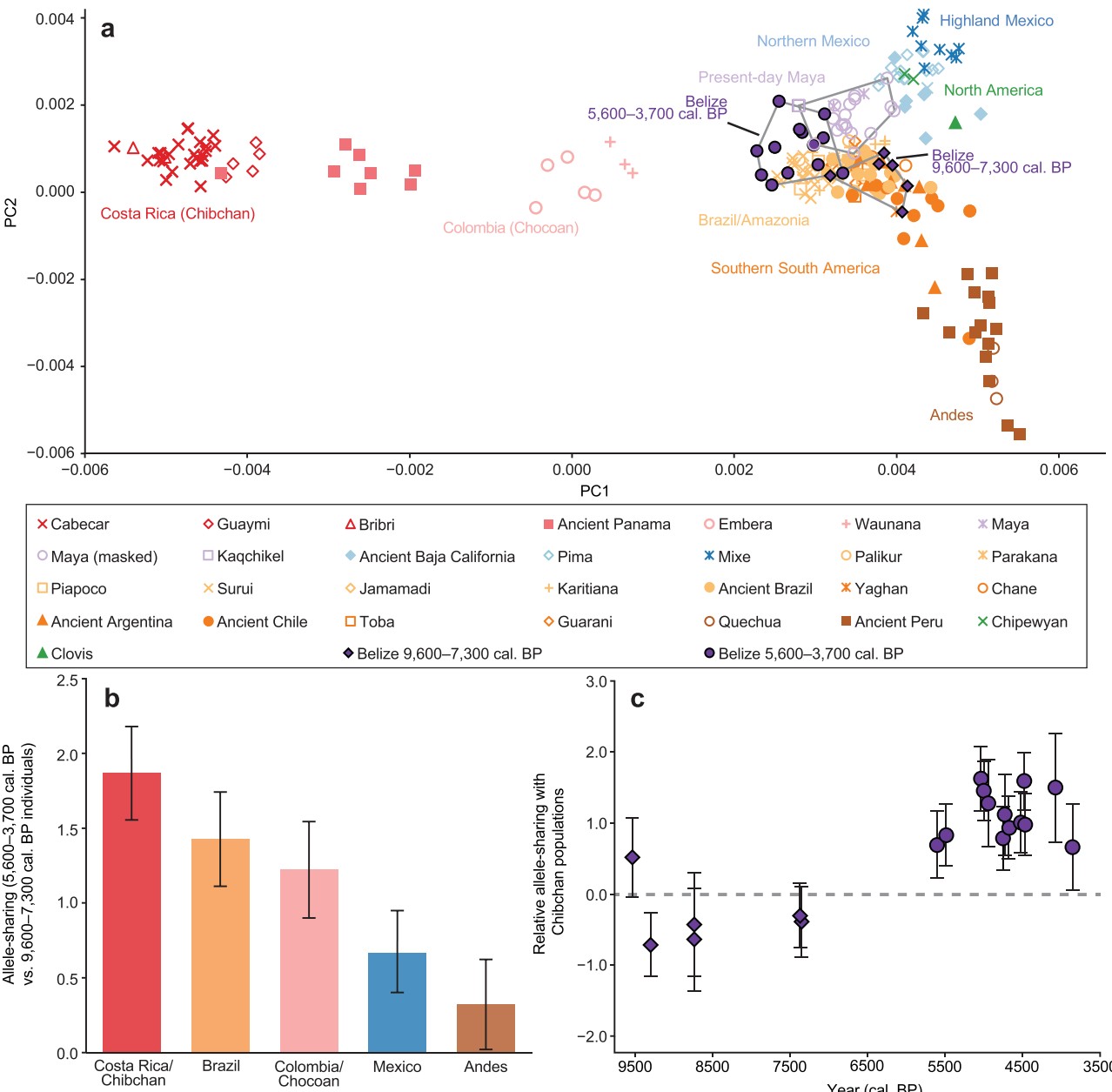

**Fig. 2 Genome-wide analyses of MHCP and ST individuals compared to present-day populations. a** PCA. Axes were computed using Maleku and Teribe (Chibchan), Zapotec (highland Mexican), and Aymara (Andean), and all individuals shown were projected (MHCP.14.2.4c was omitted due to low coverage). Alternative PCA plots can be found in Supplementary Fig. 23. **b** Allele-sharing statistics of the form $f_4$(9600–7300 cal. BP, 5600–3700 cal. BP; Outgroup, $X$) (multiplied by 1000) for present-day geographic/linguistic groupings $X$ (Chibchan: Guaymi, Maleku, and Bribri; Chocoan: Embera and Waunana; Brazil: Karitiana and Surui; Mexico: Mixe, Mixtec, and Zapotec; Andes: Aymara and Quechua). **c** Allele-sharing statistics of the form $f_4$(Aymara, Chibchan; Outgroup, $X$) (multiplied by 1000) for the ancient Belize individuals $X$ (Chibchan as in **b**). Values were computed on 305,133 SNPs. Bars show one standard error in each direction around the mean (**b** and **c**).

population transformation between the 9600–7300 cal. BP and 5600–3700 cal. BP groups based on the genome-wide data.

We explored in more detail the allele-sharing signals involving the 5600–3700 cal. BP individuals and Maya populations by computing $f_4$-statistics comparing them to other present-day groups. Consistent with the PCA results, we observe highly significant positive statistics of the form $f_4$(5600–3700 cal. BP, Maya; Central/South American, Highland Mexican) (Supplementary Table 3, Supplementary Data 8), using one Kaqchikel and two other Maya individuals (all without post-contact admixture) in the second position. These signals (combined with the excess allele-sharing between the 5600–3700 cal. BP individuals and

Maya populations) could reflect either gene flow from ancestors of highland Mexican populations (Mixe, Zapotec, and Mixtec) into the Maya lineage, or vice versa, or a combination of both. To help distinguish between these possibilities, we note that we also observe significant statistics (max $Z > 3$) with northern Mexicans (Pima Bajo) in place of highland Mexicans (Supplementary Table 5), as would be expected if the Maya individuals harbored highland Mexican-related ancestry (due to relatedness between highland and northern Mexicans). In contrast, highland Mexicans would be expected to display relatedness to Chibchan populations if they harbored Maya-related ancestry, but we do not observe such a signal among the sampled individuals

(Supplementary Table 5). Thus, while the true history may have been more complex, potentially involving admixture in both directions, our results can be explained most parsimoniously by gene flow from a source related to the ancestors of highland Mexican groups into ancestors of Maya populations sometime after 3700 cal. BP.

We used *qpAdm* to quantify mixture proportions for the admixture events described above. We obtained a good fit ($p = 0.96$) for the 5600–3700 cal. BP individuals under a two-way admixture model, with inferred proportions of 31 ± 9% ancestry related to the 9600–7300 cal. BP individuals and 69 ± 9% related to ancestors of Chibchan populations (Supplementary Table 6), providing evidence of a substantial, but not complete, ancestry shift. Given that the admixture occurred thousands of years ago, the second component would have represented a relatively early offshoot from the Chibchan-related lineage, explaining the position of the 5600–3700 cal. BP individuals in Fig. 2a (not as close to present-day Chibchan populations as to the 9600–7300 cal. BP individuals). We also obtained a good fit ($p = 0.25$) for the present-day Maya involving a mixture of ancestry related to the 5600–3700 cal. BP individuals (75 ± 10%, translating to ~52% related to ancestors of Chibchan populations and ~23% related to the 9600–7300 cal. BP individuals) and to highland Mexican populations (25 ± 10%). We replicated this result using two separate (but overlapping) sets of Maya individuals ($n = 17$ and 18) with post-contact admixture, with the non-American ancestry either masked (85 ± 7% 5600–3700 cal. BP-related, 15 ± 7% Mexican-related) or unmasked (68 ± 7% 5600–3700 cal. BP-related, 24 ± 7% Mexican-related, 7% European-related, and 1% African-related).

## Discussion

Our analysis of ancient genomes from southern Belize indicates that there were multiple phases of human migration into what became the Maya region. We confirm that the earliest sampled individuals from the region (9600–7300 cal. BP) constitute an early-splitting branch of ancestry[2], plausibly linked directly to the first southward human dispersals through the Americas during the Late Pleistocene. A gap in the skeletal record between 7300 and 5600 cal. BP is followed by more persistent burials between 5600 and 3700 cal. BP[8]. Individuals in this later cluster trace a portion of their ancestry to the earlier residents but also a portion to a source related to Chibchan populations from southern Central America and northern South America. We interpret this new ancestry as most parsimoniously explained by a south-to-north movement into the southeastern Yucatan sometime between 7300 and 5600 cal. BP. Archeological evidence throughout Central America suggests that populations were small and residentially mobile during this interval[49–51], so this movement need not have involved a large number of people moving all at once, and may instead have been an amalgamation of small populations with associated interchange of language and/or cultural knowledge. However, the net long-term effect was a substantial shift in ancestry (69 ± 9%) of local populations by ~5600 cal. BP.

This hypothesized northward movement of people related to the ancestors of Chibchan populations predates evidence for widespread maize agriculture and increases in the dietary importance of maize evident in the stable isotope record from MHCP and ST after 4,700 cal. BP[8]. Our results therefore do not support a displacement scenario of forager-horticulturalist populations by agriculturalists with an economy based on maize. However, the influx of new ancestry does correlate with evidence for regional forest disturbance and burning associated with the first small-scale maize cultivation in the region by 5600 cal. BP (Fig. 3)[25], raising the possibility that our genetic signal points to

the arrival of a new horticultural population into the region carrying maize and possibly other domesticated plants (e.g., manioc and chili peppers)[23]. Plant domestication in the Americas occurred over vast areas and involved many different species[18,39], with certain plants dispersing from Mesoamerica through Central America and South America (e.g., maize) and others moving in the opposite direction (e.g., manioc). Genetic analyses of maize have provided evidence of secondary improvements in South America[17], where it was a staple grain in coastal Peru by 6000–5000 cal. BP[20,21], followed by dispersal back into Central America[29]. Our results identify a movement of people that could have accompanied this process and been a vector for it. In particular, our results support a scenario in which Chibchan-related horticulturalists moved northward into the southeastern Yucatan carrying improved varieties of maize, and possibly also manioc and chili peppers, and mixed with local populations to create new horticultural traditions that ultimately led to more intensive forms of maize agriculture much later in time (after 4700 cal. BP)[8]. A modest increase in effective population size corresponds with the appearance of maize and these new Chibchan-related horticulturalists, consistent with the amalgamation of local and non-local populations.

The distribution and history of languages in Central America provides an independent line of evidence for our proposed historical interpretation. Chibchan is a family of 16 extant (7–8 extinct) languages spoken from northern Venezuela and Colombia to eastern Honduras (Fig. 1; Supplementary Note 3)[45]. The highest linguistic diversity of the Chibchan family occurs today in Costa Rica and Panama near the Isthmian land bridge to South America, and this is hypothesized to be the original homeland from which the languages diversified, starting roughly 5500 years ago[45]. We undertook a preliminary analysis of 25 phonologically and semantically comparable basic vocabulary items to study the linguistic evidence for interaction between early Chibchan and Mayan languages (Supplementary Tables 7, 8), of which a subset of 9 display minimally recurring *and* interlocking sound correspondences. We also focused on possible borrowings: crucially, one of the terms for maize, #ʔayma (Supplementary Tables 9, 10), diffused among several languages of northern Central America (Misumalpan, Lenkan, Xinkan)[52,53], as well as the branch of Mayan that broke off earlier than any other (Huastecan)[54]. This term is much more phonologically diverse and much more widely distributed (both across linguistic branches and geographically) in Chibchan (Supplementary Table 10) than in any other non-Chibchan language of Central America or Mexico, and it is morphologically analyzable in several Chibchan languages, including a final suffix[55]. Together, these traits support a Chibchan origin of this etymon, which could correspond to a variety of maize introduced from the south. Formal linguistic analysis over a much larger dataset will be necessary to understand the early sharing patterns (of both inheritance and diffusion) between these two language groups, which may provide further clues about the movements of material culture and people we discuss here.

These ancient DNA findings are also relevant to the ancestry of present-day Maya populations living throughout the tropical lowlands, including the Mopan and Q'eqchi' communities in southern Belize with whom we worked in the course of this project. Around 75% of Indigenous ancestry of the Maya can be traced to ancient groups living in the region between 5600–3700 cal. BP. This ancestry is in turn a combination of two components. The first was related to individuals who were buried at MHCP and ST from 9600–7300 cal. BP, who likely represented descendents of an early phase of colonization into the Americas. The second was related genetically to the ancestors of Chibchan population, and we propose was derived from a later northward

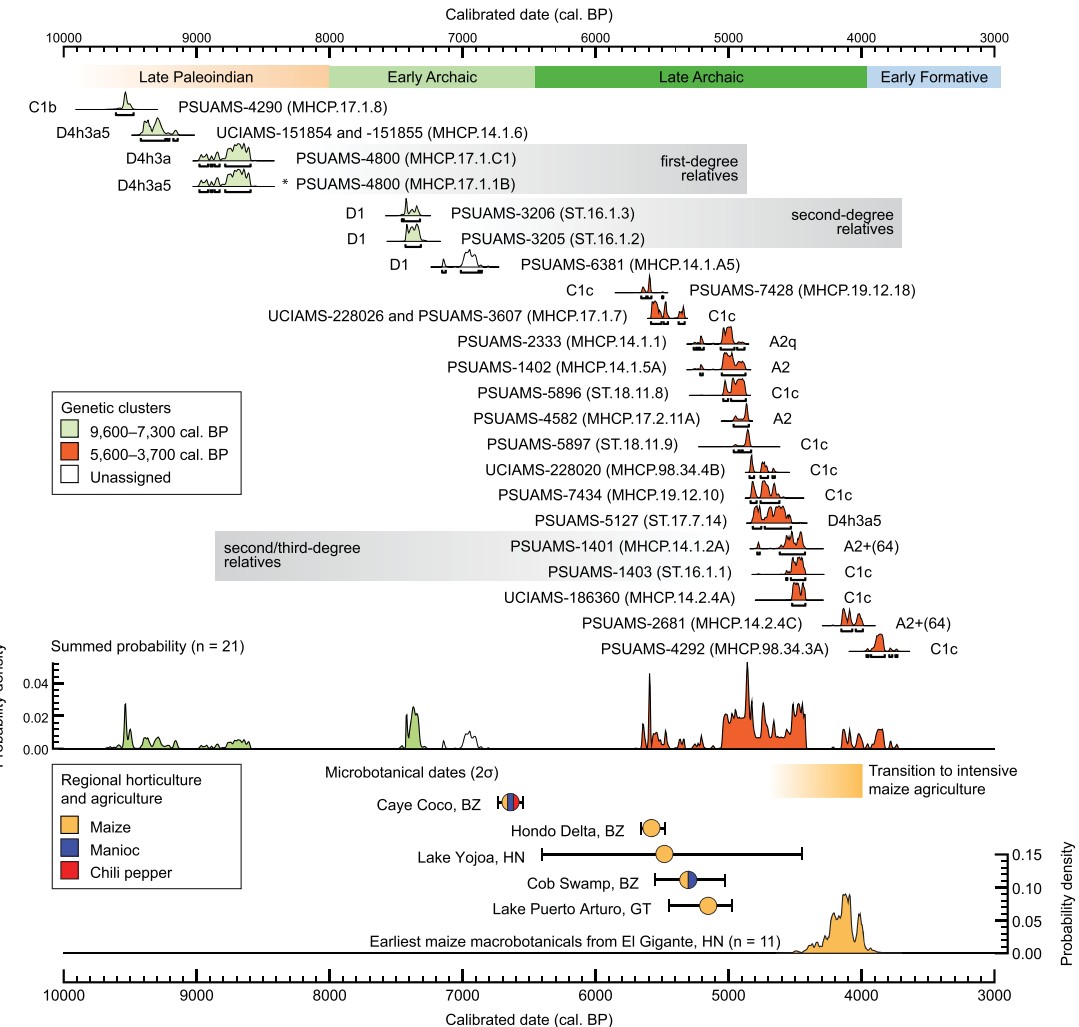

**Fig. 3 Radiocarbon dates and evidence for farming in the Maya region.** Top: dates of individuals with genetic data (individual 95.4% confidence intervals and total summed probability density; MHCP.19.12.17 [low-coverage] omitted for scale). *Date based on association with familial relative. Bottom: earliest radiocarbon dates associated with microbotanical evidence for maize, manioc, and chili peppers in the Maya region and adjacent areas at Lake Puerto Arturo, Guatemala (GT)[26]; Cob Swamp, Belize (BZ)[24]; Rio Hondo Delta[25]; Caye Coco, BZ[23]; and Lake Yojoa, Honduras (HN)[27], together with summed probability distribution of the earliest maize cobs ($n = 11$) in southeastern Mesoamerica from El Gigante rock-shelter, HN[28]. Also shown in yellow is the known transition to staple maize agriculture based on dietary stable isotope dietary data from MHCP and ST[8].

dispersal that may have been accompanied by improved varieties of maize and other domesticated plants, as well as elements of early Chibchan languages. Finally, we can model ~25% of the ancestry of present-day Mayan speakers as most closely related to highland Mexican populations. However, future research will be required to explore the full complexity of these more recent interactions in detail, including determining when such ancestry arrived in the southeastern Yucatan and how it may have related to further cultural and linguistic changes over the past several millenia.

## Methods

**Ancient DNA data generation**. We prepared ancient bone and tooth samples in clean room facilities at Harvard Medical School. We obtained powder either by sandblasting (petrous bones)[56] or drilling (other bones and teeth) after cleaning the exterior surface. From the resulting material, we extracted DNA[41,57,58] and constructed uracil-DNA-glycosylase (UDG)-treated sequencing libraries (Briggs[59]; Rohland et al.[58]; Gansauge et al.[60]) via published methods. We enriched the libraries for molecules overlapping the mitochondrial genome and ~1.2 million genome-wide target SNPs[30,43,61,62] and sequenced on Illumina NextSeq 500 and HiSeqX10 instruments.

We processed the raw sequencing data by de-multiplexing (based on library-specific barcodes), trimming adapters and barcodes, and merging reads having at least 15 bases of overlap and at most one mismatch, using in-house software tools (https://github.com/DReichLab/ADNA-Tools). Merged reads were mapped to the mitochondrial reference genome RSRS[63] and the human reference genome (version hg19)[64] using *bwa-v.0.6.1*[65]. We removed duplicate mapped reads, reads shorter than 30 bases, and reads with mapping quality scores less than 10. For use in analysis, we discarded the final 2 bases at each end of the reads (to eliminate most deamination damage-induced errors) and called pseudo-haploid genotypes at the targeted SNPs by selecting one allele at random per site.

We assessed the authenticity of the data by measuring (a) frequencies of damage-induced errors in terminal positions of sequenced molecules, (b) numbers of reads mapping to the X and Y chromosomes, (c) rates of matching of mtDNA-mapped sequences to the consensus haplogroup[43], (d) apparent heterozygosity rates at variable sites on the X chromosome in males[66], and (e) reduction in linkage disequilibrium[67]. Most individuals had minimal evidence of contamination (Supplementary Data 1, 2). For three individuals with some signals of contamination, we restricted the data to reads with damage in the final position (which should be authentic endogenous molecules; Table 1). However, we excluded two of these individuals, as well as two others, from genome-wide analyses on account of low sequencing coverage (<0.05x).

For all individuals, we determined genetic sex by comparing the fractions of sequences mapping to the X and Y chromosomes, and we searched for close family relatives by computing proportions of genome-wide alleles matching from one individual to another via in-house scripts. Mitochondrial DNA consensus

sequences were visually reviewed for ambiguities and known problematic nucleotide positions[68]. Mitochondrial haplogroups were assigned using quality scores calculated in HaploGrep 2[69] and diagnostic positions relative to the PhyloTree Build 17[68]. We determined Y-chromosome haplogroups using the tree from the International Society of Genetic Genealogy (http://www.isogg.org).

**Genetic analyses.** We merged the ancient data—including published individuals[1–5,70]—with present-day Native American genotype data[6] (using the "unadmixed" subset of individuals without evidence of post-contact admixture) and other present-day populations[71–75] (Supplementary Data 3). We used a version of the Native American data set with additional SNPs not included in the original publication due to intersections of multiple data sources, resulting in ~30% more sites, but eight fewer "unadmixed" Native American individuals as compared to the published version. Our final merged data set contained approximately 455,000 autosomal SNPs. For six populations that were not represented or had small sample sizes in ref. [6]. (Chipewyan, Maya, Mixtec, Nahua, Quechua, and Zapotec), we incorporated individuals (1–3 per population) from the Simons Genome Diversity Project (SGDP) dataset[73], although we did not use SGDP data in the $f_4$-statistics testing relatedness to present-day Maya or in qpAdm (aside from Maya themselves) to avoid possible batch-effect artifacts.

We performed PCA using smartpca[76]. We computed axes for our primary plot (Fig. 2a) using present-day Aymara (four individuals), Chibchan speakers (Maleku and Teribe, two individuals each), and Zapotec (four individuals) and projected ancient individuals (with the "lsqproject" and "shrinkmode" options) and representative present-day populations from diverse language families and geographical areas (see supplementary Fig. 23 for alternative plots). Projecting ancient individuals removes bias due to high levels of missing data, and by using a small number of populations to compute axes and projecting all present-day groups shown in the plots, we reduce the influence of population-specific genetic drift (often an important factor for populations in the axis set).

For allele-sharing analyses, we computed $f$-statistics in ADMIXTOOLS[76], with standard errors estimated by block jackknife. We used a group of 14 ancient Alaskan individuals[77] as the outgroup for $f_3$-statistics and 1000 Genomes Han Chinese (CHB)[74] as the outgroup for $f_4$-statistics, except in cases where this would result in statistics of the form $f_4$ (Ancient, Present-day; Ancient, Present-day), in which case we used the ancient Alaskans (Supplementary Data 6).

We estimated mixture proportions using the qpAdm software[61,78], with an outgroup list consisting of five Native American populations (Chipewyan, Pima, Surui, Piapoco, and Karitiana) and three eastern Eurasian groups (Chukchi, Dai, and Papuan), plus Cabecar and Waunana when modeling present-day individuals, and Russian and Dinka for the four-way model (for unmasked data from individuals with post-contact admixture). When modeling the 5600–3700 cal. BP individuals with a mixture of ancestry related to the 9600–7300 cal. BP individuals and to the ancestors of Chibchan populations, we used a combination of nine individuals to represent the second source (5 Guaymi, 2 Maleku, and 2 Bribri). Each of those populations individually yielded less precise but statistically consistent mixture proportion estimates (Supplementary Table 6). When modeling the ancestry of Maya populations, we used the 5600–3700 cal. BP individuals as one source and a combination of present-day Mixe and Zapotec as the other (with Spanish and Yoruba added as sources for the unmasked four-way model). We also obtained similar results with Mixe and Zapotec separately.

**Radiocarbon analyses.** We directly radiocarbon ($^{14}$C) dated all newly reported individuals via accelerator mass spectrometry (AMS) (Supplementary Table 11). Bone collagen yields were generally low from these depositional contexts, and multiple extractions were required for each sample to obtain datable material. In most instances extracted collagen was hydrolyzed and amino acids were purified using solid phase extraction columns (XAD amino acids[8]). Crude gelatin yields were high enough for one sample (MHCP.17.1.7) to use a modified Longin method with ultrafiltration[79]. The preservation of extracted and purified collagen or amino acid samples was evaluated using crude gelatin yields (% wt) and stable carbon and nitrogen isotope mass spectrometry (%C, %N and C/N ratios; Thermo DeltaPlus with a Costech elemental analyzer at Yale University). C/N ratios between 3.22 and 3.40 indicate that all radiocarbon dated collagen and amino acid samples were well preserved. We directly dated enamel carbonate in four samples after multiple failed attempts at extracting collagen or amino acids. Carbonate samples were chemically cleaned using published procedures[80], and sample integrity was evaluated using Fourier-transform infrared spectroscopy (FTIR, Supplementary Table 11, Supplementary Fig. 24) and stable isotope mass spectrometry[8]. To assess AMS $^{14}$C dated enamel we processed paired collagen samples from the same individuals and determined that the enamel dates were ~125–285 years younger than the paired collagen date (Supplementary Fig. 25), but we have not corrected the original dates in this analysis. After quality assurance all samples were combusted (collagen and amino acids) or hydrolyzed (carbonate) and graphitized at Penn State University (PSU) using methods described in[79]. $^{14}$C measurements were made on a modified National Electronics Corporation compact spectrometer at either PSUAMS or UCIAMS radiocarbon facilities. All dates were calibrated in OxCal version 4.4[81] using the IntCal20 curve[82] and are presented in calendar years before present (cal. BP). Multiple radiocarbon dates on the same individual were combined using the R_Combine command in OxCal.

**Reporting summary.** Further information on research design is available in the Nature Research Reporting Summary linked to this article.

## Data availability

The aligned sequences have been deposited in the European Nucleotide Archive database under accession code PRJEB49391. The processed genotype data used in analysis are available online on the *Nature Communications* website as Supplementary Data 9.

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

## Acknowledgements

Research permits were issued (to K.M.P.) by the Belize Institute of Archaeology (IA) to conduct archeological excavations, to export ancient remains to the US, and to conduct molecular analysis (2014-2018). Further permits were issued by the Belize Forest Department, in collaboration with protected areas co-managers Ya'axché Conservation Trust (to K.M.P.) to enter into and conduct research in the Bladen Nature Reserve (2014–2018). We thank the Belize Institute of Archaeology and the Belize Forest Department for permits to work in Belize and Ya'axché Conservation Trust rangers and staff for logistical support. Many thanks to Keith Hunley, James Kennett, Heather Thakar, and Barbara Voorhies for commenting on a draft of the manuscript and to Raymundo Sho, Oligario Sho, Mateo Rash, Sylvestre Rash, Jose Mes, and Julie Saul for assisting with fieldwork. Laurie Eccles kindly assisted with the radiocarbon work. We thank Ann Marie Lawson, Fatma Zalzala, Jonas Oppenheimer, Kimberly Callan, Kristin Stewardson, Matthew Ferry, Megan Michel, Nasreen Broomankhoshbacht, Nicole Adamski, and Noah Workman for ancient DNA laboratory work; Swapan Mallick, Matthew Mah, and Adam Micco for bioinformatics support; Iñigo Olalde for help with kinship analysis; Harald Ringbauer for help with ROH analysis; and Rebecca Bernardos

and Zhao Zhang for other data processing assistance. The work was funded by the Alphawood Foundation (2014-2019; K.M.P.) and National Science Foundation (SBE1632061, K.M.P.; SBE-1632144, D.J.K., B.J.C.). D.R. was supported by National Institutes of Health grants (GM100233 and HG012287), the John Templeton Foundation (grant 61220), and by the Allen Discovery Center program, a Paul G. Allen Frontiers Group advised program of the Paul G. Allen Family Foundation; D.R. is also an Investigator of the Howard Hughes Medical Institute.

## Author contributions

D.J.K., M.L., K.M.P., T.M.R., and D.R. designed the study. D.J.K., K.M.P., and D.R. supervised the study. K.M.P., M.R., W.R.T., J.R., P.L., E.M., and E.E.R. conducted field excavations. J.M., S.G., and J.J.A. provided logistical support. W.R.T., H.H.J.E., E.K.H., E.K., E.M., and L.O. performed the osteological analysis. D.J.K., T.M.R., B.J.C., and T.K.H. established the radiocarbon chronology. D.M.M. conducted the linguistic analysis. N.R. supervised ancient DNA laboratory and sequencing work. M.L. and R.J.G. analyzed genetic data. D.J.K, M.L., K.M.P., D.M.M., and D.R. wrote the paper with contributions from all authors.

## Competing interests

The authors declare no competing interests.
