## [Peer Review File · Nature Communications]

South-to-North Migration Preceded the Advent of Intensive Farming in the Maya RegionReviewers' Comments:

Reviewer #1:

Remarks to the Author:

This is the second time that I have been asked to review this paper, and I need to state from the outset but I am neither a geneticist nor a specialist in Central American archaeology or linguistics. I am reviewing this paper as an archaeologist with a general interest in the topic, particularly in the topic of early agriculture and how it spread in different parts of the world.

Many of the comments that I made in my previous review still apply here, but I detect that this paper is much improved, and easier to read than its predecessor. I regard it as well worthy of publication, partly because it is a serious attempt to combine the results from research into genetics, archaeology, and linguistics. For me, this is a very important way forward, in that many genetics papers in the past have tended to overlook important perspectives within other disciplines.

The key observation in the paper is that individuals from Belize dated between 5600 and 3700 BP are related to present-day speakers of Chibchan languages in Costa Rica and Panama, so also to modern Maya, especially in highland Guatemala, although the Maya have additional relationships with populations in western and northern Mexico that are not found amongst Chibchans. Analysis of patterns of allele sharing indicates that the population movement that best explains the observed relationships was from the Chibchan region into Belize, and not vice versa.

The suggestion that a term for maize was borrowed by Mayan from Chibchan appears to be convincing. The archaeological and stable isotope evidence for maize consumption is also clearly stated, although I was slightly puzzled by line 132, which states that agricultural populations only entered the Caribbean Islands about 1700 years ago. Unless I am misunderstanding something, the two references quoted at this point indicate a somewhat earlier date, perhaps around 2500 years ago, which is also the suggested date in the analysis of Caribbean C14 dates by Napolitano et al. in *Science Advances* 2019. However, this issue is not crucial for the paper under review.

In the discussion section of the paper, it is suggested that the south to north movement did not involve a large number of people, and that there may instead have been a gradual trickle of people that eventually reached a high point by about 5600 BP. An adjacent sentence states: "Our results therefore do not support a displacement scenario of forager-horticulturalist populations by agriculturalists with an economy based on maize."

These observations are interesting, in light of the apparent linguistic fact that the Maya do not speak languages that are transparently derived from Chibchan languages, even if there was some borrowing of a term for maize, and other linguistic influences. This makes me wonder if the movement from the south was so slow and drawn out that it never caused any language change at all in the Maya region, at least not at a full family level, even though the ultimate genetic impact appears to have climbed to more than 50% of Chibchan-related DNA in modern Mayan populations. This raises a further question – was maize farming already established in the Yucatan region by 5600 BP, as this paper seems to hint, such that the gene flow did not automatically lead to a language family replacement, but rather to a kind of amalgamation between two different populations? I am thinking here of similar situations in the western Pacific, in situations of genetic and linguistic interaction between Austronesian and Papuan speakers.

I am not suggesting that the authors need to modify their paper in light of my questions and comments, but I hope they might consider them. I am very happy to recommend this latest version of the paper for publication – it presents information that will be of considerable interest to archaeologists and linguists (not to mention geneticists, of course).

Reviewer #2:

Remarks to the Author:

The present study by Kennett et al., reports genome-wide ancient DNA data (aDNA) for 15 individuals from two Belize rock-shelters dating between 9,600-3,700 BP. The study shows that while the oldest individuals (9,600-7,300 cal. BP) descend from an Early Holocene Native American lineage with only distant relatedness to present-day Mesoamericans (including Mayan populations), more recent individuals from the same site are most likely the product of unknown human dispersal from the south related to present-day Chibchan speakers.

This results as well as other findings in this study are novel and of great interest to the fields of ancient DNA, anthropology, archaeology, among others. Therefore, results from this study advance considerably the knowledge in the aforementioned fields. The manuscript is sound, well structured, analyses are thoroughly performed undertaking the most updated approximations in the field of aDNA and population genetics, and in general inferences and conclusions are well supported by the data. While the quality of the manuscript merits its publication, there are some details which I suggest authors to address in order to improve the quality, readability and accessibility of the article.

1) Authenticity

In page 12 line 440 authors state that "authenticity of the data by measuring (a) frequencies of damage-induced errors in terminal positions of sequenced molecules, (b) numbers of reads mapping to the X and Y chromosomes, (c) rates of matching of mtDNA-mapped sequences to the consensus haplogroup, and (d) apparent heterozygosity rates at variable sites on the X chromosome in males".

While for males contamination is estimated using nuclear DNA (nuDNA) data, for females this is extrapolated from the mtDNA estimations. In all cases the deamination rate in last base has a value that will be expected for aDNA data, however this is not a quantitative evidence of lack of contamination. Moreover, male sample "I8041" doesn't have a X-chromosome contamination estimation.

Recently, the Reich lab to whom the two of the three corresponding authors and several coauthors of the study are affiliated to published ContamLD (Nakatsuka et al 2020). This method can detect nuDNA contamination by measuring the breakdown of LD due to the introduction of contaminant DNA. Authors suggest that LD patterns can be inferred from a reference sample that is known to have low estimates of contamination, and then compare such LD patterns to the LD patterns estimated in a test individual from the same/related population. For instance, males in this study for which X-chromosome contamination estimation for both the 9,600-7,300 cal. BP and 5,600-3,700 cal. BP populations can be used as reference to estimate the expected LD patterns. Then confirm such LD patterns the female individuals from the respective early or later occupation groups.

Moreover, I am also intrigued why authors chose not to use neither Contamix (Fu et al. 2013) or Schmutzi (Renaud, Slon, Duggan and Kelso 2015) both software used previously by the Reich lab.

2) Information on reference present-day populations

An additional "Supplementary Data 7" specifying the sample size of reference individuals used for reference populations, as well as the study from which each individual/populations were obtained is highly desirable.

Source articles from which data was obtained are correctly cited as described in page 12 line 460: "We merged the ancient data—including published individuals (1–5,68) —with present-day Native American genotype data 6 (using the "unadmixed" subset of individuals without evidence of post-contact admixture) and other present-day populations (69–73)".

However it is not clear which population came from which study neither the sample sizes for each one.

Regarding the data from Reich et al. 2012 it is referred that only “unadmixed” *without* evidence of post-contact admixture were incorporated into the analysis. While this information is present in Table S3 from Reich et al. 2012, it would be easier for readers if the individual ID of such individuals is also displayed in the suggested putative “Supplementary Data 7” to easily grasp the sample sample per population and the individuals used for each population. Particularly since there is already an excel file thumbnail where this information could be placed.

3) Add more published reference Maya individuals from the Yucatan peninsula to the analyses

Results from the present study show the interesting observation that present-day Maya populations unlike ancient individuals from Belize are a mixture of ancestry related to the 5,600-3,700 cal. BP individuals ($75 \pm 10\%$, translating to $\sim 54\%$ related to ancestors of Chibchan populations and $\sim 21\%$ related to the 9,600- 7,300 cal. BP individuals) and to highland Mexican populations ($25 \pm 10\%$). However, in most analysis were Maya samples are incorporated i.e. PCA, f3 and f4 *only two SGRP Maya individuals* are used as representatives of Maya genetic variation. Authors partially address this issue by showing that the ancestry profile of Maya individuals from Reich et al. 2012 study whom present some post-contact admixture overall display a similar ancestry profile to that of the two SGRP Maya individuals, despite their (low) non-Native American admixture.

Table S3 from Reich et al. 2012 provides an estimation of such non-Native American admixture in all Mayan individuals from that panel. While there are individuals with elevated non-Native American admixture ($> 10\%$), there are 16 Mayan individuals with >95 Native American ancestry: HGDP00855, HGDP00857, HGDP00859, HGDP00863, HGDP00864, Maya_4000_041700, Maya_4005_042705, Maya_4011_042711, Maya_4014_042714, Maya_4016_041716, Maya_4017_041717, Maya_4017_042717, Maya_4026_042726, Maya_4032_041732, Maya_4034_042734, Maya_4037_041737.

I strongly suggest that authors use “masked” data for those 16 Mayan individuals (or even more if possible), to project such additional Mayan individuals into the PCA and replicate f-statistic results. In addition the qpAdm results which are replicated using further Mayan individuals, results from PCA and f-statistics regarding the genetic affinities of present-day Mayan are of great relevance to the study to be only be represented by *two Mayan individuals from SGRP*. The predicted low of amount of non-Native American admixture ($< 5\%$), suggest that using masked data will no imply incorporating a considerable amount of missing data particularly when potentially several individual can be used.

4) How is the third-degree relationship inferred?

Page 12 line 450 states that “we searched for close family relatives by computing proportions of genome-wide alleles matching from one individual to another via in-house scripts”. However, to date there is no published approximation for aDNA data to characterized kinship relationships above second-degree level. Software like READ (Monroy-Kuhn 2018), among other approximations, have presented empirical and simulations to show that first and second-degree relationships can be correctly inferred using aDNA data.

However, it is not clear how third-degree connections were inferred and which threshold was used to determined such observation.

5) Missing reference aDNA samples in the study.

The study does not incorporate the ancient Panama samples from Capodiferro et al. 2021. While this samples might have been published while this manuscript was in preparation these individuals might have relevant genetic affinities to the 9,600-7,300 cal. BP and 5,600-3,700 cal. BP individuals from

Belize presented in this study. It is of great interest and potential impact to include these samples into experimental design of this study.

6) hapROH and Ne estimations

Recently hapROH (Ringbauer et al 2020a and b bioarxiv) has been used to characterize ROH tracks in precontact South American individuals (Ringbauer et al 2020). This latter study and the current submission have the same corresponding author.

However, the present study doesn't not incorporate hapROH in their analyses. Characterizing ROH in ancient individuals from 9,600-7,300 cal. BP (early) and 5,600-3,700 cal. BP (later) Belize is of interest in order to make inferences about the Ne of such populations based on their hapROH tracks. The migration described into the 5,600-3,700 cal. BP population from Chibchan-related populations could also be reflected in a higher predicted Ne for 5,600-3,700 cal. BP individuals, in comparison to the 9,600-7,300 cal. BP samples. I will recommend that authors also include hapROH in their study to further explore and confirm their results.

Reviewer #3:

Remarks to the Author:

This is a manuscript that contributes new aDNA data from an understudied time and place. It places this in context and makes inference around a fundamental issue -whether the transfer of cultivation took place with substantial migration, in this case within Mainland America. The population genomic analyses are standard for the field and seem sound and appropriate. There is a very welcome wider context and support provided by linguistic analysis (although I cannot comment on the quality of this). My judgement is that this is publishable work with enough novelty for this journal.

Some comments/criticisms follow:

I think that the reader needs to see easily the DNA data quality from each specimen, eg in a supplemental table akin to (or alongside) those with the C14 or archaeological information. This info should give an indication of the relative preservation in the samples and the quality coverage in each. This is information that is important for judging the individual data point reliabilities.

There is a detailed ethics section. However, one issue not mentioned is the value-for-destruction aspect of the work. It seems that petrous bones were powdered (completely?). Presumably these are valuable and irreplaceable specimens and partial (i.e. SNP capture not whole genome sequence) data has been gathered from them. This is therefore an incomplete genomic legacy and for a relatively modest investment (compared the price of e.g. publishing fees for this journal) this data could be expanded to whole genome sequence, leaving a more complete contribution to future studies. These data are not needed for the studies described here but the authors should address this issue, maybe by indicating future plans.

I recommend that a graph be included that plots some illustrative aspect of ancestry (eg QPGRAPH results) versus time. This is important if one wishes to convince the reader that there is a step change in ancestry probably coincident with material culture change rather than some less dramatic diffusionary process. Is there a legacy of anti-migration ideology in the Americas prehistorical community? If so all the more important to include a maximally persuasive illustration, perhaps with a timeline of other archaeological indicators.

Point-by-Point Response

We thank all of the reviewers for their comments on the manuscript. In addition to the changes outlined below, the revised version includes five more ancient individuals for whom we were able to generate data after the initial submission, increasing the sample sizes for our analyses. All additions to the main manuscript are highlighted in red, and the following figures and tables were updated or added to the supplement: Supplementary Tables 1-6 [updated], Supplementary Figure 20 [updated], Supplementary Figure 21 [new], Supplementary Figure 22 [new], Supplementary Figure 23 [updated].

Reviewer #1

Comment 1: I regard it as well worthy of publication, partly because it is a serious attempt to combine the results from research into genetics, archaeology, and linguistics. For me, this is a very important way forward, in that many genetics papers in the past have tended to overlook important perspectives within other disciplines.

Response 1: We appreciate this comment and thank the reviewer for the support.

Comment 2: The suggestion that a term for maize was borrowed by Mayan from Chibchan appears to be convincing. The archaeological and stable isotope evidence for maize consumption is also clearly stated, although I was slightly puzzled by line 132, which states that agricultural populations only entered the Caribbean Islands about 1700 years ago. Unless I am misunderstanding something, the two references quoted at this point indicate a somewhat earlier date, perhaps around 2500 years ago, which is also the suggested date in the analysis of Caribbean C14 dates by Napolitano et al. in Science Advances 2019.

Response 2: We have corrected this in the text.

Comment 3: These observations are interesting, in light of the apparent linguistic fact that the Maya do not speak languages that are transparently derived from Chibchan languages, even if there was some borrowing of a term for maize, and other linguistic influences. This makes me wonder if the movement from the south was so slow and drawn out that it never caused any language change at all in the Maya region, at least not at a full family level, even though the ultimate genetic impact appears to have climbed to more than 50% of Chibchan-related DNA in modern Mayan populations. This raises a further question – was maize farming already established in the Yucatan region by 5600 BP, as this paper seems to hint, such that the gene flow did not automatically lead to a language family replacement, but rather to a kind of amalgamation between two different populations? I am thinking here of similar situations in the western Pacific, in situations of genetic and linguistic interaction between Austronesian and Papuan speakers.

Response 3: Maize farming was not established in the Maya region prior to ~5,600 based on the available data (as summarized in the paper). In the manuscript we better highlight the fact that we do not know exactly when the population shift occurs between 7,500 and 5,600 cal BP because of a gap in the rockshelter burial record (first paragraph of discussion). Given the apparent gradual nature of subsistence change after 5,600 cal BP we do favor the idea of a slower amalgamation process as envisioned by the reviewer, but future excavations and a more continuous burial record are required to resolve this issue. In the introduction (page 3) of the manuscript we clarify the low-level use of maize, and in the first paragraph of the discussion we highlight the idea of amalgamation of small populations suggested by the reviewer.

Comment 4: I am not suggesting that the authors need to modify their paper in light of my questions and comments, but I hope they might consider them. I am very happy to recommend this latest version of the paper for publication – it presents information that will be of considerable interest to archaeologists and linguists (not to mention geneticists, of course).

Response 4: We appreciate the strong support, and these comments have helped us improve the manuscript. We have improved the discussion about the amalgamation of populations and how this may not have had a great impact on linguistic change beyond the borrowing of specific words from Chibchan, most notably the word for maize that we suggest and is worthy of further study.

Reviewer #2

Comment 1: This results as well as other findings in this study are novel and of great interest to the fields of ancient DNA, anthropology, archaeology, among others. Therefore, results from this study advance considerably the knowledge in the aforementioned fields. The manuscript is sound, well structured, analyses are thoroughly performed undertaking the most updated approximations in the field of aDNA and population genetics, and in general inferences and conclusions are well supported by the data. While the quality of the manuscript merits its publication, there are some details which I suggest authors to address in order to improve the quality, readability and accessibility of the article.

Response 1: We appreciate the positive evaluation and the suggestions, which we address point by point below.

Comment 2: In page 12 line 440 authors state that “authenticity of the data by measuring (a) frequencies of damage- induced errors in terminal positions of sequenced molecules, (b) numbers of reads mapping to the X and Y chromosomes, (c) rates of matching of mtDNA-mapped sequences to the consensus haplogroup , and (d) apparent heterozygosity rates at variable sites on the X chromosome in males”. While for males contamination is estimated using nuclear DNA (nuDNA) data, for females this is extrapolated from the mtDNA estimations. In all cases the deamination rate in last base has a value that will be expected for aDNA data, however this is not a quantitative

evidence of lack of contamination. Moreover, male sample “18041” doesn’t have a X-chromosome contamination estimation.

Recently, the Reich lab to whom the two of the three corresponding authors and several coauthors of the study are affiliated to published ContamLD (Nakatsuka et al 2020). This method can detect nuDNA contamination by measuring the breakdown of LD due to the introduction of contaminant DNA. Authors suggest that LD patterns can be inferred from a reference sample that is known to have low estimates of contamination, and then compare such LD patterns to the LD patterns estimated in a test individual from the same/related population. For instance, males in this study for which X-chromosome contamination estimation for both the 9,600-7,300 cal. BP and 5,600-3,700 cal. BP populations can be used as reference to estimate the expected LD patterns. Then confirm such LD patterns the female individuals from the respective early or later occupation groups.

Moreover, I am also intrigued why authors chose not to use neither Contamix (Fu et al. 2013) or Schmutzi (Renaud, Slon, Duggan and Kelso 2015) both software used previously by the Reich lab.

Response 2: In the revised version, we have expanded our analysis and discussion of authenticity and potential contamination, most notably through Supplementary Data 2, which contains contamLD results as well as merged mtDNA matching rates (in order to increase precision with higher coverage for individuals with multiple libraries). Other metrics are still reported in Supplementary Data 1. The contamLD estimates indicate good data quality, with upper confidence bounds of at most ~8% contamination for any individual, and almost all consistent with zero contamination. We also note that the mtDNA-based results are in fact from an implementation of contamMix (as cited in Methods).

Comment 3: Information on reference present-day populations

An additional “Supplementary Data 7” specifying the sample size of reference individuals used for reference populations, as well as the study from which each individual/populations were obtained is highly desirable.

Source articles from which data was obtained are correctly cited as described in page 12 line 460: “We merged the ancient data—including published individuals (1–5,68) — with present-day Native American genotype data 6 (using the “unadmixed” subset of individuals without evidence of post-contact admixture) and other present-day populations (69–73)”.

However it is not clear which population came from which study neither the sample sizes for each one. Regarding the data from Reich et al. 2012 it is referred that only “unadmixed” *without* evidence of post-contact admixture were incorporated into the analysis. While this information is present in Table S3 from Reich et al. 2012, it would be easier for readers if the individual ID of such individuals is also displayed in the suggested putative “Supplementary Data 7” to easily grasp the sample per

population and the individuals used for each population. Particularly since there is already an excel file thumbnail where this information could be placed.

Response 3: We have now added such a table as Supplementary Data 3.

Comment 4: Add more published reference Maya individuals from the Yucatan peninsula to the analyses

Results from the present study show the interesting observation that present-day Maya populations unlike ancient individuals from Belize are a mixture of ancestry related to the 5,600-3,700 cal. BP individuals ($75 \pm 10\%$, translating to $\sim 54\%$ related to ancestors of Chibchan populations and $\sim 21\%$ related to the 9,600- 7,300 cal. BP individuals) and to highland Mexican populations ($25 \pm 10\%$). However, in most analysis were Maya samples are incorporated i.e. PCA, f3 and f4 *only two SGDP Maya individuals* are used as representatives of Maya genetic variation. Authors partially address this issue by showing that the ancestry profile of Maya individuals from Reich et al. 2012 study whom present some post-contact admixture overall display a similar ancestry profile to that of the two SGDP Maya individuals, despite their (low) non-Native American admixture.

Table S3 from Reich et al. 2012 provides an estimation of such non-Native American admixture in all Mayan individuals from that panel. While there are individuals with elevated non-Native American admixture ($> 10\%$), there are 16 Mayan individuals with >95 Native American ancestry: HGDP00855, HGDP00857, HGDP00859, HGDP00863, HGDP00864, Maya_4000_041700, Maya_4005_042705, Maya_4011_042711, Maya_4014_042714, Maya_4016_041716, Maya_4017_041717, Maya_4017_042717, Maya_4026_042726, Maya_4032_041732, Maya_4034_042734, Maya_4037_041737.

I strongly suggest that authors use “masked” data for those 16 Mayan individuals (or even more if possible), to project such additional Mayan individuals into the PCA and replicate f-statistic results. In addition the qpAdm results which are replicated using further Mayan individuals, results from PCA and f-statistics regarding the genetic affinities of present-day Mayan are of great relevance to the study to be only be represented by *two Mayan individuals from SGDP*. The predicted low of amount of non-Native American admixture ($< 5\%$), suggest that using masked data will no imply incorporating a considerable amount of missing data particularly when potentially several individual can be used.

Response 4: We have added the masked data as suggested in both PCA and qpAdm. The results in the two analyses are similar to those obtained previously for present-day Maya populations (position in PCA and inferred ancestry proportions from qpAdm).

Comment 5: How is the third-degree relationship inferred?

Page 12 line 450 states that “we searched for close family relatives by computing proportions of genome-wide alleles matching from one individual to another via in-house

scripts". However, to date there is no published approximation for aDNA data to characterized kinship relationships above second-degree level. Software like READ (Monroy-Kuhn 2018), among other approximations, have presented empirical and simulations to show that first and second-degree relationships can be correctly inferred using aDNA data.

However, it is not clear how third-degree connections were inferred and which threshold was used to determined such observation.

Response 5: We appreciate the suggestion, and we have revised the presentation of the kinship results in Supplementary Fig 21 (with new data included as well). Expected allele-matching levels are now shown for different degrees of relatedness. We agree that third-degree kinship can be difficult to detect, and we have attempted to be cautious at that level. We now believe that the pair of individuals previously characterized as third-degree relatives should more accurately be labelled as second/third-degree (see Supplementary Fig 21), and we have made that change in the manuscript.

Comment 6: Missing reference aDNA samples in the study.

The study does not incorporate the ancient Panama samples from Capodiferro et al. 2021. While this samples might have been published while this manuscript was in preparation these individuals might have relevant genetic affinities to the 9,600-7,300 cal. BP and 5,600-3,700 cal. BP individuals from Belize presented in this study. It is of great interest and potential impact to include these samples into experimental design of this study.

Response 6: We have now incorporated the ancient Panama individuals in the relevant analyses (PCA and outgroup f_3 -statistics). As expected, they display relatedness to Chibchan populations, but we do not find evidence of particular relatedness to the ancient Belize individuals.

Comment 7: hapROH and Ne estimations

Recently hapROH (Ringbauer et al 2020a and b bioarxiv) has been used to characterize ROH tracks in precontact South American individuals (Ringbauer et al 2020). This latter study and the current submission have the same corresponding author.

However, the present study doesn't not incorporate hapROH in their analyses. Characterizing ROH in ancient individuals from 9,600-7,300 cal. BP (early) and 5,600-3,700 cal. BP (later) Belize is of interest in order to make inferences about the Ne of such populations based on their hapROH tracks. The migration described into the 5,600-3,700 cal. BP population from Chibchan-related populations could also be reflected in a higher predicted Ne for 5,600-3,700 cal. BP individuals, in comparison to the 9,600-7,300 cal. BP samples. I will recommend that authors also include hapROH in their study to further explore and confirm their results.

Response 7: In the revised version, we now report hapROH results for the individuals with sufficiently high sequencing coverage. Overall, we do not observe very strong temporal trends, but we do find evidence of higher population sizes for the later group.

Reviewer #3

Comment 1: This is a manuscript that contributes new aDNA data from an understudied time and place. It places this in context and makes inference around a fundamental issue -whether the transfer of cultivation took place with substantial migration, in this case within Mainland America. The population genomic analyses are standard for the field and seem sound and appropriate. There is a very welcome wider context and support provided by linguistic analysis (although I cannot comment on the quality of this). My judgement is that this is publishable work with enough novelty for this journal.

Response 1: We appreciate the positive comment.

Comment 2: I think that the reader needs to see easily the DNA data quality from each specimen, eg in a supplemental table akin to (or alongside) those with the C14 or archaeological information. This info should give an indication of the relative preservation in the samples and the quality coverage in each. This is information that is important for judging the individual data point reliabilities.

Response 2: A number of data quality metrics are provided in Table 1 (number of SNPs covered per individual) and Supplementary Data 1-2 (coverage, damage, and authenticity per library and per individual). We also note in the text a few instances of individuals with lower-quality data, whom we restrict to damaged sequences (which should almost all be authentic) and/or exclude from genome-wide analyses.

Comment 3: There is a detailed ethics section. However, one issue not mentioned is the value-for-destruction aspect of the work. It seems that petrous bones were powdered (completely?). Presumably these are valuable and irreplaceable specimens and partial (i.e. SNP capture not whole genome sequence) data has been gathered from them. This is therefore an incomplete genomic legacy and for a relatively modest investment (compared the price of e.g. publishing fees for this journal) this data could be expanded to whole genome sequence, leaving a more complete contribution to future studies. These data are not needed for the studies described here but the authors should address this issue, maybe by indicating future plans.

Response 3: We do indeed have plans (already partially underway) for whole-genome shotgun sequencing of individuals in this study to be used in future work (as the reviewer notes, these data are not needed to defend our primary observations and arguments in the present paper).

Comment 4: I recommend that a graph be included that plots some illustrative aspect of

ancestry (eg QPGRAPH results) versus time. This is important if one wishes to convince the reader that there is a step change in ancestry probably coincident with material culture change rather than some less dramatic diffusionary process. Is there a legacy of anti-migration ideology in the Americas prehistorical community? If so all the more important to include a maximally persuasive illustration, perhaps with a timeline of other archaeological indicators.

Response 4: We appreciate the suggestion, and we have added a figure illustrating the individual-level f -statistic results as a function of time as Figure 2C. We also display the mtDNA haplogroup distribution as part of the timeline in Figure 3. (In our view, there is indeed a history of anti-migration ideology related to the transition to agriculture in the Americas.)

Reviewers' Comments:

Reviewer #2:

Remarks to the Author:

I appreciate that Kennett et al., have addressed most of the suggestions that I made in the previous version of the manuscript. I believed that due to this revision this article have importantly increased its quality, readability and accessibility.

Nonetheless, there are still some aspects of the manuscript that I strongly recommend authors revise before its publication:

1) Previously I stated that it was not clear how third-degree connections were inferred and which threshold was used to determined such observation. Given this recommendation authors modified Supplementary Figure 21 to include the thresholds used in order to distinguish a first-degree, second-degree and third-degree or higher relationship.

However it is not clear how this thresholds are determined, and why such values are expected. For instance, at the sentence "third-degree (are expected) halfway between second-degree and unrelated" within Supplementary Figure 21 text, how does authors come to such conclusion? are they using inferences from a previous study which they should cite?

2) The methods section does not include how unmasked Maya data was analyzed for qpAdmin. Which sources and right populations were used in each case?

3) The Central/Highland Mexico Maya connection

Authors argue that the genetic connection between Central/Highland Mexico and the Maya population is better explained by migration from Central/Highland Mexico to the Yucatan peninsula (where Mayas were/are located) than in the opposite direction. Moreover, Kennett et al., mention that a result in favor of the former scenario, is that highland Mexicans do not display the relatedness signal to Chibchan populations found in both the 5,600-3,700 cal. BP individuals and present-day Maya.

However such observation could have alternative explanations given that the sample size from present-day Central/Highland Mexico is quite reduced: 1 Mixtec (SGDP), 4 Zapotec (SGDP and Reich et al 2011) and 9 Mixe (Reich et al 2011). For instance, this results could be explain by genetic structure in present-day Maya population, in the Central/Highland Mexican populations or both. So, the affinity to present-day Chibchan populations in the Central/Highland populations maybe be present but not observed with this small dataset. Also, **I am curious about why authors decide not to include the 4 Mixe genomes from the SGDP dataset which are unadmixed**

If authors want to keep the strong tone of this observation throughout the results and discussion sections, I suggest that they use additional data from other Central/Highland Mexican populations to really confirm this hypothesis. There is genotype data from present-day indigenous populations from Central/Highland Mexico in Moreno-Estrada 2013 "The genetics of Mexico recapitulates Native American substructure and affects biomedical traits", or in the recent publication by García-Ortiz 2021 "The genomic landscape of Mexican Indigenous populations brings insights into the peopling of the Americas" which has plenty Mexican indigenous populations from Central and South Mexico. Moreover, this panel has quite a few Maya and Maya-speaking populations with which all their observations could be validated and resolved in further detail.

If authors decide not to go in depth for this analysis I suggest that they tone down considerably the finding of this result both on the results and the discussion, as well as highlight the alternative scenarios that could explain such result.

Other than these three comments, I think that Kennett et al., have done a great job improving the quality, readability and accessibility of the manuscript, as previously mentioned. I will be more than

happy to recommend to publish this manuscript once this issues are addressed.

Reviewer #3:

Remarks to the Author:

My concerns have been address and I judge the MS to be acceptable.

Point-by-Point Response #2

We are grateful to the referees for the positive feedback, and appreciate their careful consideration of our revisions. Below we address four final comments made by one of the referees. These comments have helped to clarify a few remaining points and we have added new text in **blue** to the main manuscript and supplement to distinguish the changes from the alterations made during the first set of revisions in **red**.

Referee comment #2.1

Previously I stated that it was not clear how third-degree connections were inferred and which threshold was used to determine such observation. Given this recommendation authors modified Supplementary Figure 21 to include the thresholds used in order to distinguish a first-degree, second-degree and third-degree or higher relationship.

However it is not clear how these thresholds are determined, and why such values are expected. For instance, at the sentence “third-degree (are expected) halfway between second-degree and unrelated” within Supplementary Figure 21 text, how does the author come to such a conclusion? Are they using inferences from a previous study which they should cite?

Response to comment #2.1

We have added the following text and citation to Supplementary Figure 21 with a callout in the main text on page 5 of the main manuscript.

“Based on the proportion of the genome shared identical by descent (IBD) at different kinship levels, first-degree relatives are expected to have rates on average halfway between same-individual and unrelated (orange line), second-degree relatives halfway between first-degree and unrelated (brown line), and third-degree halfway between second-degree and unrelated (light blue)¹⁰.”

Reference 10 is: Monroy Kuhn, J.M., Jakobsson, M. and Günther, T., 2018. Estimating genetic kin relationships in prehistoric populations. *PLoS One*, 13(4), p.e0195491.

Referee comment #2.2

The methods section does not include how unmasked Maya data was analyzed for qpAdmin. Which sources and right populations were used in each case?

Response to comment #2.2

We have added the following text (in blue) to the methods section on pages 13-14:

“...plus Cabecar and Waunana when modeling present-day individuals, and Russian and Dinka for the four-way model (for unmasked data from individuals with post-contact admixture). When modeling the 5,600-3,700 cal. BP individuals with a mixture of ancestry related to the 9,600-7,300 cal. BP individuals and to the ancestors of Chibchan populations, we used a combination of nine individuals to represent the second source (5 Guaymi, 2 Maleku, and 2 Bribri). Each of those populations individually yielded less precise but statistically consistent mixture proportion estimates (Supplementary Table 6). When modeling the ancestry of Maya populations, we used the 5,600-3,700 cal. BP individuals as one source and a combination of present-day Mixe and Zapotec as the other (with Spanish and Yoruba added as sources for the unmasked four-way model).

Referee comment #2.3

The Central/Highland Mexico Maya connection

Authors argue that the genetic connection between Central/Highland Mexico and the Maya population is better explained by migration from Central/Highland Mexico to the Yucatan peninsula (where Mayas were/are located) than in the opposite direction. Moreover, Kennett et al., mention that a result in favor of the former scenario, is that highland Mexicans do not display the relatedness signal to Chibchan populations found in both the 5,600-3,700 cal. BP individuals and present-day Maya.

However such observation could have alternative explanations given that the sample size from present-day Central/Highland Mexico is quite reduced: 1 Mixtec (SGDP), 4 Zapotec (SGDP and Reich et al 2011) and 9 Mixe (Reich et al 2011). For instance, this results could be explain by genetic structure in present-day Maya population, in the Central/Highland Mexican populations or both. So, the affinity to present-day Chibchan populations in the Central/Highland populations maybe be present but not observed with this small dataset. Also, **I am curious about why authors decide not to include the 4 Mixe genomes from the SGDP dataset which are unadmixed**

If authors want to keep the strong tone of this observation throughout the results and discussion sections, I suggest that they use additional data from other Central/Highland Mexican populations to really confirm this hypothesis. There is genotype data from present-day indigenous populations from Central/Highland Mexico in Moreno-Estrada 2013 “The genetics of Mexico recapitulates Native American substructure and affects biomedical traits”, or in the recent publication by García-Ortiz 2021 “The genomic landscape of Mexican Indigenous populations brings insights into the peopling of the Americas” which has plenty Mexican indigenous populations from Central and South Mexico. Moreover, this panel has quite a few Maya and Maya-speaking populations with which all their observations could be validated and resolved in further detail.

If authors decide not to go in depth for this analysis I suggest that they tone down considerably the finding of this result both on the results and the discussion, as well as highlight the alternative scenarios that could explain such result.

Response to Comment #2.3

We agree it is crucial to appropriately caveat all interpretations and we have added text to our revision to do this. Our analyses do, however, allow us to confidently document a history of highland Mexican-related gene flow into the ancestry of the Maya individuals we analyzed, and after reflection and consideration of the referee's comments we do not think the new analyses suggested would change our degree of confidence in this result. We therefore have not carried out the additional analyses suggested, but have edited the manuscript to clarify our results and their implications as well as the limitations of what can be said by acknowledging possible additional complexity.

To explain our thinking here in some more detail, we note that the referee asks whether the sample we are using to represent highland Mexican ancestry is truly representative of highland Mexicans. Specifically, they write that relative to the datasets from which the highland Mexican samples came: “the sample size from present-day Central/Highland Mexico is quite reduced: 1 Mixtec (SGDP), 4 Zapotec (SGDP and Reich et al 2011) and 9 Mixe (Reich et al 2011)”

In fact, the sample we analyzed is representative of an ancestry profile that is present in a much larger number of studied highland Mexican samples. Thus, while we could of course have carried out further analyses in other datasets including those the referee cited (Moreno-Estrada et al. 2013 and García-Ortiz et al. 2021), we do not think that this would increase confidence in the finding about relatedness to diverse highland Mexicans. Co-analyzing with data from those papers would also raise technical concerns as those studies used different SNP arrays and there are some analyses where it is not good to use mixed data sources because of possible batch-effect artifacts (which we were careful to avoid for the individuals we analyzed).

To explain why the 14 highland Mexican individuals from 3 populations we analyzed are representative of a substantially larger number of individuals, we note that the Reich et al. 2012 paper (called Reich et al. 2011 by the referee) analyzed individuals from three highland Mexican populations as follows: 5 Mixtec (0 without post-colonial admixture and thus none passing filters for inclusion in the present study), 17 Mixe (9 unadmixed), and 22 from the group called in that study “Zapotec1” (from a set of homogeneously collected samples genotyped for that paper; 2 unadmixed). The clustering and PCA analyses in Reich et al. 2012 show the unadmixed samples we analyzed cluster tightly with the ones with admixture, after masking out the genomic segments derived from European and African ancestry. Thus, the samples at least for the Mixe and Zapotec are representative of the larger group (from 17 and 22 individuals respectively). Both Mixe and Zapotec show the signal of distinctive relatedness to present-day Maya but not ancient Maya reported in this study (see for example our Supplementary Table 3).

Because we wanted to include in our study additional unadmixed individuals from highly Mexico from groups with smaller sample size, we supplemented the Reich et al. 2012 dataset with data from Mallick et al. 2016 (called SGDP by the referee):

- We chose to add the 1 Mixtec individual from SGDP because this adds a sample from a third highland Mexican population thus diversifying our analyses.
- We chose to add 2 Zapotec1 individuals because we only had 2 unadmixed individuals from that group from the Reich et al. 2012 dataset. This increased our sample size from 2 to 4 Zapotec individuals.
- We chose not to add data the 3 Mixe individuals from Mallick et al. 2016 (the referee writes that there are 4 but in fact there are 3). The referee questions this: “**I am curious about why authors decide not to include the 4 Mixe genomes from the SGDP dataset which are unadmixed**”. The reasons are:
 - (a) Our Mixe sample size is quite large (n=9) from the point of view of generating statistics with small standard errors so we did not feel that it would have helped to add samples to it and we made a conscious decision that it was better to preserve the technical homogeneity of the Mixe data than to increase the sample size from 9 to 11 by adding the SGDP data to it. We continue to think this is an appropriate choice.
 - (b) There is one overlapping individual between the genetically homogeneous SGDP Mixe data and the genetically homogeneous data in Reich et al. 2012 (the individual “PT-912T / mixe0007 / B_Mixe-1”) and so we are confident that inclusion of the additional two unadmixed Mixe individuals we would have been able to add if we included the Mixe SGDP data would not have made our sample more diverse in terms of ancestry profile.
 - (c) As discussed above, the PCA and clustering analyses of the 17 individuals in Reich et al. 2012 showed that all were genetically homogeneous after correcting for post-colonial admixture so the 9 we actually analyzed are fully representative of the ancestry profile of the larger sample.

We thank the referee for their critical reading and hope we have adequately addressed the issues they raised through these replies and our revisions to the text.

Referee comment #2.4

Other than these three comments, I think that Kennett et al., have done a great job improving the quality, readability and accessibility of the manuscript, as previously mentioned. I will be more than happy to recommend to publish this manuscript once this issues are addressed.

Response to comment #2.4

We appreciate the referee’s careful reading in this and previous reviews, which has substantially improved our final manuscript.

Reviewers' Comments:

Reviewer #2:

Remarks to the Author:

I appreciate that the authors addressed the three comments I made during the second round of review. I broadly agree with their replies. I believe that the manuscript is ready to be published. Congratulations on the interesting study.

REVIEWERS' COMMENTS

Reviewer #2 (Remarks to the Author):

I appreciate that the authors addressed the three comments I made during the second round of review. I broadly agree with their replies. I believe that the manuscript is ready to be published. Congratulations on the interesting study.

We are glad the reviewer is satisfied with the way we addressed their final set of comments and thank them for their feedback on our study.